# Gender integration and female participation in scientific and health research in Zambia: a descriptive cross-sectional study protocol

Cephas Sialubanje ,[1] Phyllis Ingutu Sumbwa,[2] Nyondwa Zulu,[2] Nchimunya B Mwanza,[3] Malizgani Paul Chavula ,[3] Joseph Zulu[3]

¹School of Public Health, Levy Mwanawasa Medical University, Lusaka, Zambia
²School of Education, University of Zambia, Lusaka, Zambia
³School of Public Health, University of Zambia, Lusaka, Zambia

**Correspondence to**
Dr Cephas Sialubanje;
csialubanje@yahoo.com

## ABSTRACT

**Introduction** Despite the Zambian government making progress on prioritising gender mainstreaming, female participation in science, technology and innovation in academia, research and development is still low. This study aims to determine the integration of gender dimensions and the factors that influence female participation in science and health research in Zambia.

**Methods and analysis** We propose a descriptive cross-sectional study design employing in-depth interviews (IDIs) and survey as data collection techniques. Twenty schools offering science-based programmes will be purposively selected from University of Zambia (UNZA), Copperbelt University, Mulungushi University and Kwame Nkrumah University. In addition, two research institutions, Tropical Disease Research Centre and Mount Makulu Agricultural Research Station, will be included. Survey respondents will include a random sample of 1389 academic and research staff from the selected schools. A total of 30 IDIs will be conducted with staff and heads of selected schools and research institutions. Data collection will be conducted over a 12-month period. Before commencement of data collection, in-depth literature search and record review on gender dimensions in science and health research will be conducted to gain insight into the subject matter and inform research instrument design. Survey data and IDIs will be collected using a structured paper-based questionnaire and semistructured interview guide, respectively. Descriptive statistics will be computed to summarise respondents' characteristics. Bivariate analysis ($\chi^2$ test and independent t-test) and multivariate regression analysis will be conducted to test the association and identify factors influencing female participation in science and health research (adjusted ORs, p<0.05). An inductive approach will be used to analyse qualitative data using NVivo. Survey and IDI will be corroborated.

**Ethics and dissemination** This study involves human participants and was approved by UNZA Biomedical Research Ethics Committee (UNZABREC; UNZA BREC 1674-2022). Participants gave informed consent to participate in the study before taking part. Study findings will be disseminated through a written report, stakeholder meetings and publication in a peer-reviewed international journal.

## STRENGTHS AND LIMITATIONS OF THE STUDY

⇒ Use of a mixed-methods cross-sectional design employing in-depth literature review, survey and in-depth interview techniques increases the internal validity of the study.

⇒ Use of probability proportional to size and random sampling techniques to select schools and study participants, respectively, minimises bias due to selection bias.

⇒ Training of data collectors minimises measurement error.

⇒ Cross-sectional design of the study does not allow for causal inference.

⇒ Recruiting study participants from only four public institutions of higher learning and two research institutions may introduce selection bias and limit validity of study findings.

⇒ Selection of all the 16 schools offering science-based programmes from only two public institutions in may introduce selection bias.

## INTRODUCTION
### Background

Gender mainstreaming is a public policy concept of assessing different implications of any public policy action, legislation and programmes on people of different genders.[1] The concept was first proposed at the 1985 third World Conference on Women in Nairobi, Kenya.[1] It has been adopted into the United Nations' (UN's) development community as a globally accepted strategy for promoting gender equality[2] and ensuring that all UN member states achieve the 50% gender parity—through empowering all women and girls, ending all forms of discrimination against women and girls everywhere, and ensuring women's full and effective participation and equal opportunities at all levels of decision making.[3] Gender equality ensures that women have the same opportunities in life as men, including the ability to participate

BMJ

in the public sphere.[4] It also ensures that gender perspectives and goals are central to all activities including policy development, research, advocacy/dialogue, legislation, resource allocation, planning, implementation, and monitoring of programmes and projects. Gender equality also ensures equitable distribution of benefits from the development process for boys, girls, men and women. It is at the heart of strategies and interventions driving the active and meaningful participation of both women and men in the development agenda and ensures consistent empowerment of women and girls.[4] It is a basic requirement for measuring progress on sustainable development goals (SDGs) and constitutes a key aspect within the principle of leaving no one behind, without which the SDGs cannot be achieved.[5] Both gender equality and women's empowerment are integral to realising the 2030 Agenda for Sustainable Development[6] and all the UN SGDs[6] which were agreed on in the UN Country Team System-wide Action Plan. Thus, the gender dimension of science and technology has become an increasingly important and topical issue worldwide.

Globally, countries are increasingly recognizing the importance of supporting the study of science and technology in higher institutions of learning for both men and women.[7] Consequently, enrolment of women at the tertiary level has increased over the past few decades, with women now approaching 50% of the total number of tertiary students worldwide. The greatest gains in absolute terms have been made in developing countries. However, there are wide variations at the country and region levels.[7]

Nevertheless, women and girls around the world are still excluded from participation in science and technology, and gaps remain in the area of women researchers and scientists in top managerial positions throughout the world.[7–9] In southern Africa, less than 30% of women participate in research; the proportion varies across the countries, ranging from 8.7% in Democratic Republic of the Congo to 44.9% in South Africa and 47% in Eswatini. The most preferred areas of research focus are natural sciences (31%) and agricultural and veterinary sciences (18.7%). Although the European Union for Research and Innovation (2022) report[10] shows that 57% of the European Union (EU) tertiary graduates are women, less than 4% of female students take degrees in engineering, manufacturing, construction and information, communication and technology (ICT) which are predominantly chosen by male students; only 23% of women participate in medical science research, 19% in natural sciences research and 4% in agricultural sciences research.

Zambia is one of the UN Member States that have adopted the gender mainstreaming concept and developed a national gender policy.[11] The policy focuses on attainment of gender equality in the development process and providing for equal opportunities for women and men to actively participate and contribute to their fullest ability. It also focuses on equitable benefits from national development.

Although the Zambian government has made significant progress in adopting the policy[12] and increasing female enrolment in secondary and higher education as well as their representation in decision-making positions,[13–15] female representation and participation in science, technology and innovation (STI) is still low and far from achieving the UN 50% gender parity target. Reports and studies conducted in Zambia[13–15] reveal unequal gender representation in STI with women continuing to lag behind their male counterparts; their access and retention in tertiary education remains a challenge. For example, the Ministry of Gender and Child Development report[15] shows that women only comprised 39.1% of the combined total enrolment in the four Zambian public universities (University of Zambia (UNZA), Copperbelt University, Mulungushi University and Kwame Nkrumah University).

Reasons for the low female participation in this field are not well understood.[16–19] Several factors, including poverty, lack of education, and aspects of their legal, institutional, political and cultural environments, have been shown to affect women from participating in science and innovation. However, there is dearth of published literature on this subject; most published studies relate to developed countries and regions such as the USA and Europe.[20] The few published studies focus on ICT[21–26] in secondary and higher institutions of learning, which are not applicable to STI in academia, research and development. Thus, the actual gender context with regard to integration of gender dynamics and the issues affecting female participation in science and health research in the country are not well understood despite the growing recognition of the importance of gender equality in this area. A study is therefore required to answer the following questions in order to bridge the knowledge gaps on this subject:

1. To what extent is gender mainstreaming integrated in science and health research?
2. What is the proportion of women participating in science and health research in Zambia?
3. What factors influence female participation in science and health research in the country?

### Objectives

This study aims to explore the integration of gender dimensions and determine the predictors of female participation in scientific and health research in the country. The specific objectives are:

1. Explore integration of gender policy in academic and research institutions.
2. Explore female participation in science and health research activities.
3. Identify the factors that affect female participation in science and health research and development.
4. Determine the predictors of female participation in scientific and health research.

This knowledge is important for informing policy and programming for the advancement of gender equality,

equal opportunities and participation in science, health research and development in order to promote quality scientific and health outcomes and technological excellence. Incorporating gender dimensions and deliberately encouraging women to participate in science and health research is crucial in realising the key role that women can play in the development and advancement of science, and their commitment to inclusive science and research development. Gender equality in science and research is not only a matter of fairness but is important for enhancing the recruitment of the most talented individuals, irrespective of gender, and for tapping a partially unexploited resource.[27 28] It is a way to promote scientific and technological excellence rather than just improving opportunities for women.[29 30] A more inclusive workforce is likely to be more innovative and productive than one which is less so.[31] Moreover, having scientists and researchers with diverse backgrounds, interests and cultures assures better scientific research and the best use of those results.[32] Mainstreaming gender in science and health research is crucially important for various reasons. First, women are the main driving labour force in most African countries, making a vital contribution to the economy. Empirical evidence[33] shows that if women had equal access to productive resources such as science, technology, research or financial services, the gender productivity gap would almost disappear and African countries, including Zambia, could significantly increase their production by just closing the gender gap across the economic, academic and scientific divide. Second, increasing women's participation in STI has the potential to lead to higher Gross domestic product (GDP) growth through increases in higher-skilled women in the labour force and women's employment in high-productivity sectors. Gender equality and empowerment of women to participate in science and health research can also boost women's role as leaders in the field.[33 34]

## METHODS
### Study design
We propose to conduct a descriptive mixed-methods cross-sectional study design employing in-depth interviews (IDIs) and survey as data collection techniques. Use of qualitative and quantitative data collection and analysis techniques will make the study comprehensive and allow for triangulation and corroboration of the findings. This will in turn increase the internal validity of the findings.[35 36]

### Study setting
The study will be conducted at four public institutions of higher learning in Zambia (UNZA, Copperbelt University, Kwame Nkrumah University and Mulungushi University) and two public research institutions (Tropical Disease Research Centre ((TDRC) and Mount Makulu Agricultural Research Station). In addition, one Science-Granting Council (SGC) and the Higher Education Loans and Scholarship Board (HESB), two government ministries (Gender and Higher Education) and United Nations agencies (UNICEF and UNESCO) will be included in the study. UNZA is located in Lusaka, the capital city of Zambia. It is the largest and oldest public university in the country. It was founded in 1965 and established by the Act of Parliament No. 66 of 1965. The first intake of students took place on 17 March 1966. It offers a number of academic programmes at diploma, undergraduate and postgraduate levels. The university has 13 faculties including Agricultural Sciences, Education, Engineering, Graduate School of Business, Health Sciences, Law, Medicine, Mining, Public Health, Natural Sciences, Nursing Sciences, Social Sciences and Humanities, Veterinary Medicine. It also has the directorate of research and graduate studies (DRGS). The Copperbelt University is the second largest and oldest public university. It is located in Kitwe, Zambia and has 12 schools including: School of the Built Environment, School of Business, Dag Hammarskjold Institute for Peace and Conflict Resolution, Directorate of Distance Education and Open Learning, School of Engineering, School of Graduate Studies, School of Mathematics and Natural Science, School of Medicine, School of Mines and Mineral Resources, School of Natural Resources, School of Information and Communication Technology, and School of Humanities. Mulungushi University and Kwame Nkrumah University are located in Kabwe, the capital of Central Province. TDRC is a public health research institution located in Ndola, Copperbelt Province. Mount Makulu Agricultural Research Station is a public agricultural research institution located in Chilanga District, Lusaka Province—approximately 20 km from the national capital, Lusaka. HESB is a public institution located in Lusaka. It provides loans and scholarships to students in to pursue higher education both in Zambia and outside.

A total of 16 schools offering science-related and health-related programmes will be purposively selected from the selected institutions. Ten schools offering science-related and health-related programmes will be purposively selected from UNZA, namely: Medicine, Public Health, Nursing Sciences, Health Sciences, Veterinary Medicine, Mines, Engineering, Natural Sciences, Agricultural Sciences, Directorate of Research and Postgraduate Studies. Seven schools offering science-related and health-related programmes at Copperbelt University will be included in the study: School of Engineering, School of Mathematics and Natural Science, School of Medicine, School of Mines and Mineral Resources, School of Natural Resources, School of Information and Communication Technology, and School of Graduate Studies. In addition, two schools from Mulungushi University and one from Nkrumah University will be included in the study. Two research institutions, TDRC from Ndola Zambia and Mount Makulu Agricultural Research Station in Chilanga, will be included in the study.

## Participants and sampling process

Survey participants will comprise a systematically selected sample of 1389 male and female academic and research staff from the selected schools . The academic and research institutions, science granting council and scholarship board have been selected because they are public and are involved in science and health research.

To be eligible, survey participants will be:

► Lecturers and researchers at any academic rank (from the lowest level to full professor).
► Those having worked in the selected institution for at least 3 months.

IDI participants will include staff from the selected schools and research institutions. To be eligible, IDI participants will be:

► Administrative staff from academic institutions (Registrar or Assistant Registrar, School Dean or Assistant Dean, Head of Department or Section Head, director of DRGS).
► Administrative staff from research institutions (head or assistant head of selected research institution, section head of research institution).
► Academic staff at all ranks.
► Technical staff from research institutions.

Survey participants will be selected using a stratified sampling technique. First, the number of staff from each school will be obtained. To ensure that all schools are proportionately represented, probability proportional to size will be conducted to determine how many participants will be selected from each school. Next, a list of all the lecturers in each selected school will be compiled to serve as a sampling frame. To ensure inclusion of both male and female participants from various academic and administrative ranks and also to ensure that they have an equal chance of being selected, information on the sex (male and female) of potential respondents will be obtained from the administrative managers and separate lists will be prepared for each sex. The sampling frame will also include other variables: qualification (first degree, masters and PhD), position and rank in the organisation (from lowest rank to the full professor), administrative rank (ordinary lecturer, unit head, department health, Dean, Registrar, etc). In order to be comprehensive, a checklist will be prepared by the research team in consultation with administrative staff from the respective organisation. Each potential participant will be assigned a unique number in the sampling frame. A random number generator will be used to select participants from the list. This process will be done for each institution and school.

Purposive sampling will be used to recruit IDI participants from the selected institutions. In order to compare and contrast the respondents' views, both male and female participants will be included. We plan to conduct a total of 30 IDIs in both academic and research institutions. However, the actual number of IDIs to be conducted will be determined by the point of saturation.[37–39]

In order to get variability in responses, and to compare and contrast the similarities and differences in participant perspectives on the subject, efforts will be made to ensure that both male and female respondents are included in the IDIs and questionnaire survey. Efforts will also be made to have a mix of survey participants from the lowest (lecturer 3) to the highest (full professor) ranks.

## Sample size estimation

In order to detect the difference in participation in scientific and health research between women and men, we will use the power approach to estimate the sample size. The power approach is used to compute the sample required to determine the minimum difference between two or more groups in comparative studies.[40 41] In order to use this approach, the following assumptions will be made:

► Power =80%
► 95% CI
► $\alpha=0.05$
► Groups = two (male and female)
► Sampling ratio =1
► Minimum detectable difference =15.0%
► Margin of error =5%.

Inputting these assumptions into the online advanced power and sample size calculator[42] gives a mimimum sample size of *604* per group or 1208 for the total sample. Allowing for a 15% non-response rate (NRR) gives a total sample size of *1389 (or 695 for each group) survey respondents* required to have enough power to measure the proportion of women participating in STI and carry out multivariate analysis to determine the predictors of female participation in STI with 95% confidence.

## Variables

The study variables will include the outcome and independent variables as follows:

### Outcome variable

Female participation in science and health research and development.

### Independent variables

A. Demographic information: age, sex, number of children.
B. Field of study.
C. Academic rank.
D. Attitude towards scientific and health-related research.
E. Treatment at the workplace.
F. Workplace culture.

## Data sources and collection techniques

In order to gain insight into the gender integration issues and inform design of data collection instruments, we propose a review of existing documents and policies from the UN agencies, government ministries of gender and higher education, HESB, science granting councils, and academic and research institutions. The proposed search will include: (A) Enrolment and graduation data from the four selected academic institutions, (B) Administrative data on staffing profile in the selected schools; (C)

Data on both external and internal research grants in the academic institutions, (D) Data on local grants from science granting councils, (E) Review of national and institutional documents on gender policy and integration.

To determine the factors influencing gender integration and female participation in science and health research, a survey using a structured questionnaire will be conducted among a randomly selected sample of 1389 academic staff from the selected schools. To qualitatively explore the respondents' perspectives and gain insight into gender integration and female participation in science and research, IDIs will be conducted with administrative and academic staff from the academic and research institutions as shown in online supplemental table 1.

## Data collection procedures

Data collection will be conducted over a 12-month period. IDIs will be conducted by a pair of trained research assistants under the supervision of the research team members. Research assistants will be selected from the Master of Public Health students in the School of Public Health at UNZA. Only students with experience in qualitative research will be selected. Survey questionnaires will be emailed to the respondents by the members of the research team.

Prior to data collection, research assistants will undergo training for a total of 5 days. The training will be conducted by the principal investigators (PIs) and the co-investigators, and will be in two phases: theory and fieldwork. Phase I will include theory in research and will take 3 days. Topics will include: (1) Basic principles of qualitative and quantitative research, (2) Purpose and objectives of the study, (3) Data collection and interviewing techniques, (4) Research ethics, and (5) Informed consent in human subjects' research. Phase II will be field practicals and will include pretesting of the data collection tools and consent form, and will take 2 days, after which the instruments will be revised based on the collected and analysed data and feedback from the research assistants. Each IDI and questionnaire is expected to last between 30 min and 45 min.

We plan to conduct record review first followed by IDIs, and then use insights from the literature review and qualitative study to revise and improve the questionnaire for the quantitative survey.

## Data collection instruments

A paper-based, standard, structured and pretested questionnaire (online supplemental material 2) will be used to collect survey data. The questionnaire has been developed based on the existing literature and in consultation with the research team. The questionnaire will also be informed by the findings from the IDIs. In order to to increase its validity, the questionnaire will be pretested and revised during the practical training of data collectors. To make it easier to administer and measure constructs, questions on attitude and other constructs will be based on a 5-point Likert Scale. To make it easy to collect survey data, the questionnaire together with the consent form will be emailed to the selected participants. Instructions will be provided on how to fill in and return the questionnaire.

IDIs will be collected using a pretested interview guide (online supplemental material 3) that has been prepared by the research team. The same interview guide will be used for all the participants from the study sites. The interview guide will be developed based on specific objectives: (A) Perspectives on integration of gender in science and research, (B) Participation/involvement in science and research, (C) Factors influencing female participation in science and research. The survey questionnaire will also have the same sections but will include an additional section on sociodemographic and economic characteristics. A checklist (online supplemental material 4) will be used for record review. The questionnaire, interview guide and checklist will all be in English. This is because the study participants (academic and research staff) speak English. Before each IDI, respondent sociodemographic data including age, sex, marital status, occupation, level of education and level of monthly income will be collected using a short questionnaire. Interviews will be recorded on digital audio recorders and notes will be taken by research assistants. The data collection tools will be pretested and revised before being used in the field. The data collection tools are summarised in online supplemental table 2.

## Bias

Possible biases in the study include selection bias, measurement error and confounding due to various background characteristics. To minimise selection bias, systematic stratified sampling will be used to select participants from the line lists maintained in the study sites. In addition, a group of research team members will work together to select the study sample. Other measures to increase validity of the study will include : (A) Estimation of the sample size using scientific methods to minimise random error; (B) Pretesting and revision of data collection instruments translated into the local languages (Tonga, Bemba and Nyanja) to minimise measurement error, (C) Sampling procedures will be followed to minimise selection bias; (D) Training of research assistants to minimise measurement error. In addition, research assistants will work under supervision from the field supervisors; (E) Survey data will be entered and analysed by an independent and competent person to reduce data entry errors; (E) Coding and analysis of qualitative data will be done by two research team members to increase internal validity. Confounding due to background variables will be controlled for during data analysis. During fieldwork, the team will ensure quality in the data collection process by a) ensuring that questionnaires are completely filled in, a) taking extensive notes during key informant interviews, b) having evening meetings to review the day's activities and challenges.

## Statistical methods and data analysis

Quantitative/survey data will be analysed using SPSS V.21 (IBM, Armonk, New York, USA). Descriptive statistics and frequencies will be used to summarise survey data. Categorical data will be presented using percentages, quantitative data will be presented using the mean and SD (or median and IQR). Before running bivariate and multivariate regression analyses, factor analysis will be conducted on each psychosocial measure using principal axis factoring as an extraction method and oblimin rotation. Next, inspection of the scree plot (that is, a plot which displays the eigenvalues associated with a component or factor in descending order vs the number of the component or factor) will be done and sum measures will be created with an eigenvalue score of 1.0 or higher and those items with factor loadings of 0.4 or higher. Reliability test for internal consistency using Cronbach's α will be conducted on each of the items. Items that show strong internal consistency (Cronbach's α 0.70 or r >0.40) will be combined and averaged into one variable.

Bivariate analysis will be done using Pearson's correlation (for continuous variables) and $\chi^2$ test (for categorucal variables) to test the association between the independent and outcome variables. Multivariate logistic regression analysis will be conducted to identify predictors of female participation in science and research (r>0.10, p<0.05).

Qualitative data analysis will be done using an inductive approach which ensures that subthemes are derived from the predetermined themes by content analysis and grouping all similar statements made with respect to particular themes. The voice recordings from the key informant interviews will be transcribed by the research assistants. After verification of accuracy in transcription, each transcript will then be thoroughly read by one research assistant while the other one will be listening to the corresponding voice recording. Each transcript will then be compared with the handwritten field notes that the research assistants will prepare during the IDIs. After proof-reading and making corrections, the transcripts will be saved on a password-protected file, in a computer kept by the PI, for safety. The Word documents will then be imported into NVivo V.11 MAC for coding and analysis. The exported data will then be coded using a codebook and guiding framework, and the categories and key subthemes will be identified.

## Patient and public involvement

The study design was determined by the call for application for a research grant on gender dimensions in STI under the strategic research fund. Thus, the participants and the public were not directly involved in the conceptualisation and design of the study. Nevertheless, selection of the 16 schools from the two institutions was done in consultation with the authorities from the ministries of higher education and gender. Further, recruitment of study participants will be done in collaboration with the respective school deans and their assistants. To inform the institutions about the study, a letter will be written to the registrars and heads of the research institutions. Next, meetings will be held with the two respective school deans and the heads of the two research institutions to discuss the study and recruitment of survey and IDI participants. Finally, a report will be written and shared with key stakeholders, including the funding organisation and ministries of higher education and gender. In addition, a dissemination meeting will be held in Lusaka to share the results with the two ministries and heads of the participating schools and institutions.

## Potential risks and mitigation

This study poses minimal risk to study participants. Research assistants will not administer the questionnaire; questionnaires will be delivered to the respondents' office for self-administration. Completed questionnaires will be sealed in an envelope for research assistants to collect. Participation in the study may cause some discomfort from answering certain questions, particularly if the respondent had a negative gender-related experience. To reduce the risk of disclosure of personal or sensitive information, data collectors will be trained to ensure that they advise participants not to share anything that they are not comfortable with; and that they do not have to respond to any question unless they feel comfortable doing so. Since the questionnaire will be self-administered, respondents will have minimal risk of COVID-19 infection.

The study may expose IDI participants and study team members to COVID-19 infection. Biosafety measures will be taken during data collection including consistent wearing of face masks, social distancing and conducting virtual interviews to minimise the risk of infection to the research team and study participants. In addition, data collectors and study participants will be encouraged to consistently use face masks and hand sanitisers before and after each interview. Used face masks will be disposed of according to the health guidelines. Data collectors are also trained to minimise any potential discomfort or harm to all participants during all study activities to the greatest extent possible. The study team will reduce any waiting by participants by scheduling appointments during times convenient to participants and interviews are kept to as short a time as possible. To ensure data quality, weekly research meetings will be held to ensure effective and efficient execution of the project.

## Potential benefits

There are no direct individual benefits to participating in the study. If the proposed study is effectively implemented, the findings will be disseminated among the stakeholders. The findings are likely to lead to design of evidence-informed strategies to improve female participation in science and research.

## Participant confidentiality

Confidentiality of collected data from study participants will be assured throughout the study. The interview will be carried out in the participant's office or place of

choice. Data collection will not be conducted until we can confirm that the time and location are acceptable to the participants. On completion of IDI and survey, all survey data will be entered in Excel and stored on a secure pass-code protected computer kept by the PI. IDI audio recordings will be transcribed and stored on the same computer. Both survey and IDI data will be de-identified and the linking file with identifiable data and basic demographics will be stored in a separate file. Only the PI and some identified research teams will have access to the identifiable data. All analyses will be conducted on de-identified data.

## Ethics and dissemination

Ethical clearance and approval were obtained from the UNZA Biomedical Research Ethics Committee (UNZA BREC 1674–2022). Permission to conduct the study was granted by the National Health Research Authority. Before being recruited into the study, the purpose, process and duration of the study, benefits and associated risks, privacy and confidentiality will be explained to the respondents. Data collectors will explain that there are no direct individual benefits to participating in the study. They will also explain potential risks the study may pose to study participants and that participation in the study is not likely to pose significant risk. To reduce the risk of disclosure of personal or sensitive information, data collectors will be trained to ensure that participants are helped not to share anything that they are not comfortable with; and that they do not have to respond to any question unless they feel comfortable doing so. Participants will also be told that they are free to stop the discussion at any time if they need to. Moreover, participation may expose the study participants and study team to COVID-19 infection. To minimise the risk of infection to the research team and study participants, biosafety measures will be taken during data collection including social distancing and conducting virtual interviews. In addition, data collectors and study participants will be encouraged to consistently use face masks and hand sanitisers before and after each interview. Used face masks will be disposed of according to the health guidelines. Data collectors will also be trained to minimise to the greatest extent possible any potential discomfort or harm to the participants during all study activities. The study team will minimise any waiting by participants by scheduling appointments during times convenient to participants and ensuring that interviews are kept to as short a time as possible. Those who accept to participate will be asked to sign the consent form (online supplemental material 6) and be enlisted in the study. Written informed consent will be obtained from each respondent. Those who cannot read or write will be asked to mark with an 'X'. The consent form will be translated into the local language. Participants aged below 18 years will require both parental consent and assent. Privacy of participants and confidentiality of collected data will be assured throughout the study. Interviews will be carried out in participants' private homes or at the

participant's place of choice. Data collection will not be conducted until we confirm that the location is acceptable to the participants. To ensure data quality, weekly research meetings will be held to ensure effective and efficient execution of the project. On completion of each data collection phase, all files will be stored on a secure pass-code protected computer kept by the PI. At the end of the data collection process, data will be de-identified and the linking file with identifiable data and basic demographics will be stored in a separate file. Only the PI and some identified research team members will have access to the identifiable data. All analyses will be conducted on de-identified data.

If the study is successful, the findings will be disseminated among the stakeholders. A report will be written at the end of the project and shared with key stakeholders, including the funding organisation and Ministry of Health. In addition, study findings will be disseminated to key stakeholders in Zambia, then through open-access peer-reviewed journals, websites and international conferences.

## Limitations

This study has several limitations. First, the study will only include 1389 survey respondents and 30 IDI participants from four academic and two research institutions. The study findings may not be generalisable to other institutions. Moreover, schools connected to the arts and humanities have been excluded as these often have a relatively high proportion of women; the actual proportion of women in these schools is not known. In addition, these schools have been excluded because our project is on science and health research. To make the study findings more representative, a large-scale study is needed involving more academic and research institutions. The purpose of the current study is to gain insight on gender integration in science and health research, describe female participation in science and health research, and identify the associated factors. Second, non-response may affect the external validity of the study. To mitigate this, we have allowed for a 15% NRR. Furthermore, limited access to research data from the selected institutions may affect the quality of the study. Finally, the cross-sectional study will not allow for conclusion on causality. An experimental study is needed to allow for conclusion on causation of low female participation in science and research.

Despite these limitations, we believe that the study is likely going to contribute to the scientific body of knowledge on the integration of gender mainstreaming in science and research. The findings will also provide insight on female participation in science and research and the factors which affect their participation. Currently, no such study has been conducted in the country and information on this subject is scant.

## Costs and payments

There is no cash payment provided to participants for any portion of the study. Participants volunteer the time taken

to complete this survey. As a small token of appreciation for their time and opportunity costs, IDI participants will receive a bottle of water or soft drink, in line with UNZA BREC procedures.

**Contributors**  CS led the study design, drafting of the study protocol and implementation of the study, and drafted this manuscript. JMZ and MPC contributed to the study protocol. PIS, NBM and NZ contributed to the development of the data collection instruments, enlisted the study sample, and coordinated the data collection process. PIS and NZ contributed to the revision of the manuscript. All authors read and approved the manuscript.

**Funding**  This work was supported by the National Science and Technology Council (grant number NSTC 101/6/8; online supplementary material 6), as part of gender dimensions in science, technology and innovation under the strategic research fund. The funders had no influence on the design and implementation of the study, and drafting of this manuscript.

**Competing interests**  None declared.

**Patient and public involvement**  Patients and/or the public were involved in the design, or conduct, or reporting, or dissemination plans of this research. Refer to the Methods section for further details.

**Patient consent for publication**  Not applicable.

**Provenance and peer review**  Not commissioned; externally peer reviewed.

**Data availability statement**  Data are available upon reasonable request.

**ORCID iDs**
Cephas Sialubanje http://orcid.org/0000-0002-9077-1436
Malizgani Paul Chavula http://orcid.org/0000-0003-1189-7194

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

4  ASTI. African women in agricultural research and development. n.d. Available: asti.cgiar.org/gender
5  United Nations Sustainable Development Group. Gender equality and empowerment of women: UNCT-SWAP gender equality scorecard. 2018. Available: unsdg.un.org/resources/unct-swap-gender-equality-scorecard
6  United Nations. Sustainable development goals. Transforming our world: the 2030 agenda for sustainable development. 2015. Available: sustainabledevelopment.un.org/post2015/transformingourworld
7  UNESCO. UNESCO science report: the race against time for smarter development. Download the Report | 2021 Science Report; 2021. Available: unesco.org
8  Gender in science innovation, innovation, technology and engineering. n.d. Available: genderinsite.net/about/who-we-are
9  UNESCO. Sustainable development goals for natural sciences. n.d. Available: en.unesco.org/sustainabledevelopmentgoalsfornaturalsciences
10  European Union. Science, research and innovation performance of the EU 2022 report: building a sustainable future in uncertain times; 2022. Available: europa.eu
11  UNESCO. Science, technology and innovation policy. n.d. Available: www.unesco.org/new/en/natural-sciences/science-technology/sti-systems-and-governance/
12  JICA. Country gender profile: Zambia final report; 2016. Available: www.jica.go.jp/english/our_work/thematic_issues/gender/background/c8h0vm0000anjqj6-att/zambia_2016.pdf
13  Ministry of Gender and Child Development and Central Statistics Office. Gender status report 2012-2014. n.d. Available: www.zamstats.gov.zm/phocadownload/Gender/Gender%20Status%20Report%202012-2014%20290616.pdf
14  UNESCO. UNESCO science report: towards 2030: measuring progress towards sustainable development goal 9. n.d. Available: en.unesco.org/unesco_science_report/sdg9
15  Ministry of Gender and Child Development. National gender policy. 2014. Available: genderlinks.org.za/srhrs/zambia-national-gender-policy-2014/
16  Begeny CT, Ryan MK, Moss-Racusin CA, *et al*. In some professions, women have become well represented, yet gender bias persists-perpetuated by those who think it is not happening. *Sci Adv* 2020;6:eaba7814.
17  Cheryan S, Ziegler SA, Montoya AK, *et al*. Why are some stem fields more gender balanced than others? *Psychol Bull* 2017;143:1–35.
18  National Academies of Sciences, Engineering, and Medicine. *Promising practices for addressing the underrepresentation of women in science, engineering, and medicine: opening doors*. National Academies Press (US), 2020.
19  Hill C, Corbett C, St Rose A. *Why so few? Women in science, technology, engineering, and mathematics*. 1111 Sixteenth Street NW, Washington, DC 20036: American Association of University Women, 2010.
20  Castillo R, Grazzi M, Tacsir E. *Women in science and technology. What does the literature say?* Inter-American Development Bank, 2014. Available: publications.iadb.org/en/women-science-and-technology-what-does-literature-say
21  Lee H, Pollitzer E. *Gender in science and innovation as component of inclusive socioeconomic growth*. Portia Limited, 2016.
22  Nicia G, Luisa S. A gender-based assessment of science, technology and innovation ecosystem in Mozambique. *AFJRD* 2020;5.
23  Lardies CA, Dryding D, Logan C. *Gains and gaps: perceptions and experiences of gender in Africa*. 2019.
24  Kunda D, Chembe C, Mukupa G. Factors that influence zambian higher education lecturer's attitude towards integrating icts in teaching and research. *J Technol Sci Educ* 2018;8:360.
25  UN. Mainstreaming a gender perspective in science, technology and innovation policy. United Nations commission on the status of women. 2011. Available: un.org
26  Nyanja N, Musonda E. A review of the ICT subject implementation in schools: a perspective of Lusaka Province (Zambia). *Educ Inf Technol* 2020;25:1109–27.
27  Hamusankwa M, Munsaka E. *Gendered experiences of female engineering students in selected public universities in Zambia*.
28  Kipsoi DrEJ, Chang'ach DrJK, Sang HC. Challenges facing adoption of information communication technology (ICT) in educational management in schools in Kenya. *JSR* 2012;3:18–28.
29  Morsy H. Mainstreaming gender in african policymaking: ensuring no voice is unheard. 2019. Available: reliefweb.int/report/world/mainstreaming-gender-african-policymaking-ensuring-no-voice-unheard
30  Zavale NC. Expansion versus contribution of higher education in Africa: university–industry linkages in Mozambique from companies' perspective. *Sci Public Policy* 2018;45:645–60.
31  United Nations. Envision 2030 goal 9: industry, innovation, and infrastructure. n.d. Available: www.un.org/development/desa/disabilities/envision2030-goal9.html
32  UNESCO. Gender equality guidelines for UNESCO publications. n.d. Available: en.unesco.org/system/files/ge_guidelines_for_publications_-_annex_5.pdf
33  UN Women. Gender mainstreaming. n.d. Available: www.un.org/womenwatch/osagi/gendermainstreaming.htm
34  UNICEF. Gender main streaming assessment. 2007. Available: www.unicef.org/evaldatabase/files/EO_2007_Gender_Mainstreaming_Report.pdf
35  King N, Horrocks C. *Interviews in qualitative research*. 1st ed. SAGE Publications ltd, 2010: 25–41.

36 Ary D, JacobsL, SorensenC, *et al*. *Introduction to research in education, 9th ed*. Wadsworth, CA: Cengage Learning, 2013.

37 Massar K, Sialubanje C, Maltagliati I, *et al*. Exploring the perceived effectiveness of applied Theater as a maternal health promotion tool in rural Zambia. *Qual Health Res* 2018;28:1933–43.

38 Sialubanje C, Sitali DC, Mukumbuta N, *et al*. Perspectives on factors influencing transmission of COVID-19 in Zambia: a qualitative study of health workers and community members. *BMJ Open* 2022;12:e057589.

39 Sialubanje C, Mukumbuta N, Ng'andu M, *et al*. Perspectives on the COVID-19 vaccine uptake: a qualitative study of community members and health workers in Zambia. *BMJ Open* 2022;12:e058028.

40 Whitley W, Ball J. Statistics review 4: sample size calculations. 2002. Available: ccforum.com/content/6/4/335

41 Jones SR, Carley S, Harrison M. An introduction to power and sample size estimation. *Emerg Med J* 2003;20:453–8.

42 Power and sample size calculator. Sample size calculator - calculate power & sample size for one-sample, two-sample and k-sample experiments. n.d. Available: gigacalculator.com

