## [Reviewer comments · BMJ Open]

ARTICLE DETAILS

TITLE (PROVISIONAL)	Gender integration and female participation in scientific and health research in Zambia: A descriptive cross-sectional study protocol
AUTHORS	Sialubanje, Cephas; Sumbwa, Phyllis; Zulu, Nyondwa; Mwanza, Nchimunya; Chavula, malizgani; Zulu, Joseph

VERSION 1 – REVIEW

REVIEWER	Begeny, Christopher University of Exeter
REVIEW RETURNED	04-Jul-2022

GENERAL COMMENTS	This manuscript (protocol) outlines plans for a project that aims to better understand the extent to which, and potential barriers to, gender integration and female participation in scientific and health research in Zambia. It proposes using a mixed methods, cross-sectional study that includes record review, in-depth interviews (IDs), and a (quantitative) survey. I should mention that my feedback is from the perspective of a quantitative social psychologist (who does research in STEM contexts), and so I will focus my review on the quantitative aspects of the proposed project, along with some of the more general methodological aspects of the project. Overall, I thought the manuscript was well written, and the protocol was fairly well-developed. However, there were areas where the rationale for certain decisions was not clear to me, and there was an assortment of other little issues (e.g., use of causal language when there are no plans to conduct research that will enable tests of causality). I also thought the authors could better position themselves to thoroughly and effectively capture the nature of (if not gender disparities in) female participation in science and health research – along with predictors of this – by considering the inclusion of some additional measures in their quant. survey. All comments below are focally for the authors to consider, in case any of it seems useful for generating a more robust and insightful project. As to whether any or all of these comments should be addressed through revision to the manuscript itself, I trust the handling editor's discretion on this. QUANTITATIVE SURVEY 1. One of the authors' stated aims is to understand the factors that influence female participation in science and health research in Zambia. I wonder if some of the insights from social psychological
---

literature would be helpful, both in terms of capturing relevant predictors and outcome variables (granted the vast majority of that work is rooted in US and European countries...a sore limitation of that work to be sure).

In part, this might include thinking about 'female participation' in a more nuanced way. Beyond whether someone does (vs. does not) participate/have a position in science and health research, or what (categorical) role they occupy, you might consider assessing the *degree to which* they engage in their science or health research role. I think will lead to a lot of important insight around gender disparities in female participation (i.e., beyond disparities in numerical representation, are there disparities in levels of job satisfaction, career motivations, etc etc...gender disparities that are important not only in their own right but also because they can be predictors of job retention/dropout and so in this way also shed light on issues of numerical representation).

For example, you could measure levels of career ambition/motivation to pursue higher goals and positions, levels of psychological (dis)engagement or burnout at work, levels of work-related stress, desire to leave the area/profession, level of identification with the area/profession (e.g., how important it is to one's self-concept, level of pride they take in being a part of the science and health research community), and/or levels of job satisfaction – not in yes/no terms, but in grades (e.g., 1 – 7, Not At All Satisfied – Extremely Satisfied). You might also consider assessing some relevant self-report behaviors (e.g., frequency of engaging in extra-role behaviors at work, or similarly organizational citizenship behaviors).

In terms of predictor variables, the authors might consider including some measures that assess how individuals are treated in the workplace (e.g., *how often* individuals experience gender-based discrimination, how often individuals experience fair treatment, distinctive treatment, etc.). They might also consider a measure of workplace culture (e.g., to what extent does it seem fair, equitable, inclusive), or access to role models.

If they wanted to assess some key mechanisms (between these predictors and outcome variables), they might include a measure of perceived inclusion/belonging, and/or perceived value/worth (e.g., within one's college/faculty, within the science and research community), sense of work-life balance, expectations of success, etc. There is existing literature on all of the aforementioned constructs, which the authors could draw on to get existing measures and a more detailed understanding of the relevant theory/processes around them.

Overall, if the authors choose to include more of the types of constructs, measured on Likert-type scales (e.g., 1-7), I do think this would help them capture a lot of important differences (and gender disparities!) in the level of engagement/participation in science and health research, along with key predictors of these disparities. More generally, I think this would position the authors to better address and speak to the fact that gender disparities can exist in a variety of forms – separate from the issue of numerical representation (and continuing to be an issue, even when numerical representation is not an issue anymore! see, e.g., Begeny, C. T., Ryan, M. K., Moss-Racusin, C. A., & Ravetz, G.

(2020). In some professions, women have become well represented, yet gender bias persists—Perpetuated by those who think it is not happening. *Science Advances*, 6(26), eaba7814.

Potentially useful reviews on (sociocultural predictors of) gender disparities in participation, engagement and retention in various STEM-related professions/fields:

Hill, C., Corbett, C., & St Rose, A. (2010). *Why so few? Women in science, technology, engineering, and mathematics*. American Association of University Women. 1111 Sixteenth Street NW, Washington, DC 20036.

National Academies of Sciences, Engineering, and Medicine. (2020). *Promising practices for addressing the underrepresentation of women in science, engineering, and medicine: Opening doors*. National Academies Press.

Cheryan, S., Ziegler, S. A., Montoya, A. K., & Jiang, L. (2017). Why are some STEM fields more gender balanced than others?. *Psychological bulletin*, 143(1), 1.

OTHER POINTS

2. It appears that IDIs are planned exclusively for individuals in more leadership-type roles. Why so?

Perhaps this selection criteria makes sense (e.g., given the questions that are planned for the IDIs) – but is it also possible that some important perspectives and insight will be missed by only interviewing individuals who are in positions of relative power?

Might the view from those “at the top” be a relatively narrow view?

So to say, might individuals in leadership positions tend to have (developed) perspectives that differ from individuals who are not in leadership positions? And if so, why only gather in-depth information from those who are in more established positions of power? Why not aim for a mix of participants from the lowest (lecturer 3) to highest rank (full prof), as with the survey data?

All to say, I would love to hear more about the rationale for using this inclusion criteria for IDIs (i.e., only conducting IDIs among individuals with relatively higher status and/or power).

3. “To allow for triangulation and validation of findings, both qualitative and quantitative studies will be conducted simultaneously.”

I struggle to understand the rationale for conducting the qualitative and quantitative studies simultaneously. Would it not be more useful to conduct the qualitative studies first, and then use insights from that to revise and improve the information collected in the quant study? If conducted simultaneously, what will the authors do if they glean some really important and unanticipated insights from their qualitative work (and yet be unable to follow up on it quantitatively)?

4. "Allowing for a 10% nonresponse rate (NRR)..."

What is the supporting evidence or rationale for anticipating a 10% NRR? I worry that this is a very optimistic estimate (i.e., in reality, the NRR will be a lot higher), and if so the authors will end up with a particularly underpowered study, which could make it very difficult to interpret any results as reliable or robust – also making it hard to publish this work in a reputable journal.

More generally, the target sample size seems quite low for the variety of analyses planned. More information about the authors' power analyses would be helpful (e.g., does it account for multiple tests and Type I error rate? What is the desired power to detect effects?). Similarly, more information about the authors' planned analyses would be helpful (e.g., what types of multivariate regression analyses are they planning to conduct, and how many of them? Given the variety of faculties/colleges/fields of research they plan to examine, are there any plans to test for similarities/differences across them [e.g., via use of multilevel modelling]?)

5. "Items that show strong internal consistency (Cronbach's alpha 0.60 or $r > 0.40$) will be combined and averaged into one variable based on factor analysis."

First, a cronbach's alphas of .60 is quite low (below conventionally accepted standards in psych research, at least), and so I wonder about the authors' rationale for using this threshold.

Second, the proposed plan here is a bit confusing. An inter-item correlation (reflected in a cronbach's alpha value) is not itself a factor analysis. Do the authors plan to just look at cronbach's alpha values, or do they plan to conduct factor analyses? And if they do plan to conduct factor analyses, what types of factor analyses do they plan to conduct: EFAs? CFAs? If CFAs, do they plan to compare results to any alternative factor models? If EFAs, what are their plans for these analyses (e.g., rotation method, approach for determining factor extraction, etc.; for a review of best practices, see: Costello, A. B., & Osborne, J. (2005). Best practices in exploratory factor analysis: Four recommendations for getting the most from your analysis. *Practical assessment, research, and evaluation*, 10(1), 7)?

Overall, I would recommend that they do conduct factor analyses, because a high inter-item correlation does not mean that those items constitute a conceptually meaningful (single, unidimensional) construct.

6. "In addition, two research institutions Tropical disease research centre (TDRC) from Ndola Zambia, and Mount Makulu Agricultural Research station in Chilanga were included in the study."

Why so? I don't see this as problematic necessarily, but what is the rationale for including these institutions?

	7. What is the difference between STI and STEMM, and why the focus on STI as opposed to STEMM? From the selection of schools/faculties being recruited from, it appears that the authors include Engineering- and Mathematics-related schools/faculties (e.g., Engineering at UNZA, School of Mathematics and Natural Science at Copperbelt). 8. "A longitudinal study is needed to allow for conclusion causation of low female participation in science and research" Longitudinal research can be informative for understanding how processes operate over time (e.g., whether different constructs systematically co-vary across time), but it does not allow one to test for nor claim causality. Experiments are key to testing and claiming causality.
--	---

REVIEWER	Coates, Laura Oxford Brookes University Faculty of Health and Life Sciences, NDORMS
REVIEW RETURNED	16-Dec-2022

GENERAL COMMENTS	This is a very interesting protocol paper describing a mixed-methods study to examine gender balance in scientific and health research in Zambia. The strengths and limitations of the proposed study are well described and the methods are clearly stated. 1. Will any data be collected from the other institutions? Even if the study is focusing on the two public universities, basic data could potentially be sought from other institutions to see how the basic gender balance compares. 2. In similar questionnaires, we have asked about part-time or less-than-full-time working. Is this an issue worth exploring in the study? (this may be less relevant in the Universities being studied).
---

REVIEWER	Herbert, Rachel Elsevier BV, International Center for the Study of Research
REVIEW RETURNED	06-Jan-2023

GENERAL COMMENTS	The aims of the project are admirable and will resonate with wider ambitions both in Zambia and far beyond to support gender equality in all areas of society. Providing evidence on the details of representation among researchers and HEI faculty and on the opinions around the barriers and opportunities experienced by women will support future policy reviews. In the main, the authors lay out an approach which I believe will elicit useful and impactful results which will, most importantly, facilitate discussion in the workshops and through the generation of the report. Spelling and grammar errors are present throughout the paper. I would advise careful review of this throughout the entire paper. In the main, the paper is understandable despite these errors. I have picked out those that I noticed them and included details here. These errors have not played a part in my recommendation. Page 2, line 48: I would advise additional keywords: gender discrimination, gender equality, Zambia, representation. And add a common after 'research'.
--

	Page 4. Line 14: Change 'eempowering' to 'empowering' Page 4, line 31: The acronym SDG is used on line 31, but is laid out in full on the later line 36: these should be switched so that the first instance of SDG is spelt out fully. Where spelt out, the words should use title case. Page 4, line 42: Delete the first (repeated) instance of 'gender' Page 4, line 53: Delete the underscore after the square bracket Page 5, line 8: The latest data on female representation at the HEIs is now quite out of date (2011). In the context of the aims of the aims of this project (see question 2 at the end of this section), this seems noteworthy: does this represent a limitation and challenge for those working in Zambia trying to understand the policy impact and provide evidence for future reviews/updates? Page 5, line 17: There are more up to date numbers from UNESCO here: https://www.unesco.org/reports/science/2021/en/dataviz/share-women-researchers-radial (underlying data is also available). Page 5, line 23: change 'countries' to 'countries and regions' Page 5, line 47: change 'the country' to 'in the country' Page 5, line 49: Objectives 1 and 2 here are not particularly specific, and so it will be difficult to identify whether those are achieved. Can you expand/be more specific with your plans here. Page 6, line 6: change 'science, health research and development' to 'science and health research and development' Page 6, line 24: Do you have a source for the statement on the labor force? Page 6, line 25: Do you have a source for the evidence mentioned? Page 7, line 14: Earlier (page 5, line 23), you state that there have been studies in the field of ICT, and yet Copperbelt's School of ICT is excluded here: what is the reasoning? And further, while the focus of the project is on Science and Health, it might be noted that the decision to exclude schools connected to the arts and humanities have been excluded as these often have a relatively high proportion of women. This should be noted as a limitation of the study. Page 8, line 18: delete duplicate period at end of sentence. Page 8, line 22: how is sex data on the staff being obtained? Page 8, line 43: change 'esmiate' to 'estimate' Page 8, line 45: change 'resspondnets' to 'respondents' Page 8, line 46: Can you expand on this decision? You state that the proportion of females is not known, yet you note earlier (page 8, line 22) that you will be obtained this characteristic as you obtain the staff counts. And the proportion is known as of 2012 for Zambia: see UNESCO reporting 25.26% for Zambia in 2012 here: https://www.unesco.org/reports/science/2021/en/dataviz/share-women-researchers-radial. Page 10, line 13: change 'IDis' to 'IDIs' Page 10, line 15: change 'researck' to 'research' Page 10, line 17: missing word after 'Master of Public' according to the acronym Page 10, line 22: the topics included cover the technical approaches required, but has consideration been given to the potentially sensitive nature of responses? For example, how to handle discussions around harassment or other improper conduct? Page 10, line 35: change 'quantittative' to 'quantitative' Page 14, line 36: period missing from end of sentence Page 22, line 16, question 4: It may be relevant to ask whether the partner is also employed by a higher education / research institution.
--	--

	Page 22, line 42, question 8: I find this an ambiguous question and am unclear on what the result would provide. Page 22, line 47, question 9: option 1 of the answers is not a person. I'd also advise a option to say 'no-one'. The question might be rephrased to 'Did anyone influence your choice of career?' If yes, Who? (multiple options). Page 23, line 17-47, questions 10-14: These all fit better with the introductory questions (1-7). If you move those, there would then also be a better flow if question 16 follows 10. Page 16, line 58-60, question 16: might you also add (1) the citation impact of prior work and (2) the value of grants awarded Page 24, line 5, question 17: The question appears incomplete. And a binary answer doesn't seem appropriate. Could you change it to 'To what extent are you satisfied by your current role', and then offer a scale from Very Satisfied to Unsatisfied Page 24, line 11, question 18: For clarity, you could delete 'science and' Page 25, line 39, question 21: Add 'Other...' Page 25, line 10-20, questions 25-26: I have strong reservations here. The structure of the options for Qu 26 risk leading the answers and are not respectful of the experience of harassment in the workplace. I would advise redressing the approach here entirely, and supplementing the questionnaire with useful information for a respondent having to discuss this by, for example, providing a helpline and by being clear on the confidential nature of the survey. In addition, the authors must define what they mean by harassment. Then a better approach would be to ask firstly, have you had any experience of harassment at your current or a former workplace (no, yes – I have, yes – I have witnessed it, yes, I have heard second hand reports). Then a follow up might ask to what extent was this a factor if they decided to leave a workplace (answers: it was the driving factor, it was a contributing factor, it was not a factor, not applicable – I have not left). Page 26-27, lines 56-8, questions 44-46: Again, these questions fit better with the initial introductory questions, where the authors also ask about children. Page 26, line 9-34, questions 33-44: It is notable here that the authors ask about various policies, but not about the effects felt or the opportunities (if any) that these bring out. This may be well served by the IDIs, but I do perceive the questionnaire as focusing on the barriers in the main and providing quite little opportunity to bring to light positive changes. Page 27, line 10-20, questions 47-48: This is a very sudden shift in the line of questioning. I would advise sections/thematic headers to be added throughout the questionnaire to help the respondent. Furthermore, these two questions are ambiguous: can the authors confirm what type of accomplishments and give examples of the types of financial and non-financial incentives they are referring to? Page 27, line 54-57, questions 55-56: Might the value of the grants also be of interest to the authors, as this can play a critical part in what research teams can accomplish? Page 27, line 59, question 57: This is another abrupt shift in the line of questioning and a section header would help. Furthermore, this question is open to interpretation: the authors should rephrase the question to better explore this idea. For example, this could be interpreted as 'all else being equal, can women experience as much success as women in research?' or 'in your workplace, can women experience as much [career] success as men?' Asking why might be beneficial to the project's objectives.
--	---

	Page 28, line 6, question 58: I advise separating the question around field and job: the answers could be very different. Asking why might be beneficial to the project's objectives.
--	---

VERSION 1 – AUTHOR RESPONSE

Reviewer # 1

1. One of the authors' stated aims is to understand the factors that influence female participation in science and health research in Zambia. I wonder if some of the insights from social psychological literature would be helpful, both in terms of capturing relevant predictors and outcome variables (granted the vast majority of that work is rooted in US and European countries).

Response: We appreciate the reviewer's comment. We have now included the specific objective: To determine the predictors of female participation in scientific and health research (page 5). We have maintained the outcome variable as: Female participation in science and health research and development (page 9)

We have also taken into consideration the reviewer's suggestions on how to frame the outcome and predictor variables. We have also included the suggested references in our reference list. Our questionnaire already has the suggested measures: levels of career ambition/motivation to pursue higher goals and positions, levels of psychological (dis)engagement or burnout at work, levels of work-related stress, desire to leave the area/profession. The questionnaire is constructed on a 5-point Likert scale (see attached questionnaire).

Response: We appreciate the reviewer's suggestion on the predictors. We have revised this section and included other predictors as suggested (page 9). We will edit the questionnaire accordingly.

2. It appears that IDIs are planned exclusively for individuals in more leadership-type roles. Why so?

Response: We appreciate the observation by the reviewer. We have edited the selection criteria for IDI participants to include other participants who are not in positions of authority (page 8)

3. To allow for triangulation and validation of findings, both qualitative and quantitative studies will be conducted simultaneously. I struggle to understand the rationale for conducting the qualitative and quantitative studies simultaneously.

Response: We appreciate the reviewer's comment. We have edited this section to read, "we plan to conduct the IDIs first, and then use insights from qualitative study to revise and improve the questionnaire for the quantitative survey" (page 11).

4. Allowing for a 10% nonresponse rate (NRR)..." What is the supporting evidence or rationale for anticipating a 10% NRR? I worry that this is a very optimistic estimate (i.e., in reality, the NRR will be a lot higher), and if so the authors will end up with a particularly underpowered study, which could make it very difficult to interpret any results as reliable or robust – also making it hard to publish this work in a reputable journal.

More generally, the target sample size seems quite low for the variety of analyses planned. More information about the authors' power analyses would be helpful (e.g., does it account for multiple tests and Type I error rate? What is the desired power to detect effects?). Similarly, more information about the authors' planned analyses would be helpful (e.g., what types of multivariate regression analyses are they planning to conduct, and how many of them? Given then variety of faculties/colleges/fields of research they plan to examine, are there any plans to test for similarities/differences across them [e.g., via use of multilevel modelling]?)

Response: We appreciate these observations. We have edited the whole section on sample size estimation and recalculated the sample size using the formula for comparative studies (male and

female participants) the sample size has now increased to 1,389. The sample size was arrived at using the following assumptions: 80% power; sampling ratio of 1; 95% confidence interval; alpha of 0.05; margin of error at 5%. We have also allowed for a 15% non-response rate according to the acceptable standards of a minimum of 85% response rate (page 9).

We have described in detail the types of analysis we hope to carry out (see Statistical methods and data analysis on page 12)

5. "Items that show strong internal consistency (Cronbach's alpha 0.60 or $r > 0.40$) will be combined and averaged into one variable based on factor analysis."

Second, the proposed plan here is a bit confusing. An inter-item correlation (reflected in a cronbach's alpha value) is not itself a factor analysis. Do the authors plan to just look at cronbach's alpha values, or do they plan to conduct factor analyses? And if they do plan to conduct factor analyses, what types of factor analyses do they plan to conduct: EFAs? CFAs? If CFAs, do they plan to compare results to any alternative factor models? If EFAs, what are their plans for these analyses (e.g., rotation method, approach for determining factor extraction, etc.; for a review of best practices, see: Costello, A. B., & Osborne, J. (2005). Best practices in exploratory factor analysis: Four recommendations for getting the most from your analysis. *Practical assessment, research, and evaluation*, 10(1), 7)?

Response:

We have edited the whole section and explained how factor analysis and reliability test will be conducted. In short, principal factor analysis will be to extract components using oblimin rotation. Reliability test for internal consistency using Cronbach's alpha will be conducted on each of the items. Items that show strong internal consistency (Cronbach's alpha 0.70 or $r > 0.40$) will be combined and averaged into one variable (page 12 & 13).

6. "In addition, two research institutions Tropical disease research centre (TDRC) from Ndola Zambia, and Mount Makulu Agricultural Research station in Chilanga were included in the study."

Why so? I don't see this as problematic necessarily, but what is the rationale for including these institutions?

Response: We appreciate this concern and corrected the error in the document. We plan to include a total of six institutions: four academic and two research institutions as well as one Science Granting Council, two government ministries (Gender and Higher Education) and Un agencies (UNICEF and UNESCO) into the study (page 7)

7. What is the difference between STI and STEMM, and why the focus on STI as opposed to STEMM?

From the selection of schools/faculties being recruited from, it appears that the authors include Engineering- and Mathematics-related schools/faculties (e.g., Engineering at UNZA, School of Mathematics and Natural Science at Copperbelt).

Response: STI stands for science, technology and innovation; STEM stands for science, technology, engineering and mathematics. We decided to focus on STI because our expertise is in health research and our interest is in female participation in scientific and health-related research and not in STEM (page 5).

8. "A longitudinal study is needed to allow for conclusion causation of low female participation in science and research". Longitudinal research can be informative for understanding how processes operate over time (e.g., whether different constructs systematically co-vary across time), but it does not allow one to test for nor claim causality. Experiments are key to testing and claiming causality.

Response: We appreciate the comment. We have edited this section to read: An experimental study is needed to allow for conclusion on causation of low female participation in science and research (page 16 & 17).

Reviewer: 2

1. Will any data be collected from the other institutions? Even if the study is focusing on the two public universities, basic data could potentially be sought from other institutions to see how the basic gender

balance compares.

Response: We appreciate the observation on the number of institutions. We have corrected this section and explained that we plan to include a total of six institutions: four academic and two research institutions as well as one Science Granting Council, two government ministries (Gender and Higher Education) and Un agencies (UNICEF and UNESCO) into the study (page 7)

2. In similar questionnaires, we have asked about part-time or less-than-full-time working. Is this an issue worth exploring in the study? (this may be less relevant in the Universities being studied).

Response: We appreciate the comment. In our questionnaire we have a question on whether someone is full time or not (see question 12)

Reviewer 3

1. Spelling and grammar errors are present throughout the paper. I would advise careful review of this throughout the entire paper. In the main, the paper is understandable despite these errors. I have picked out those that I noticed them and included details here. These errors have not played a part in my recommendation.

Response: We appreciate the reviewer's observation. We have proofread the whole document and corrected all the grammatical and typo errors.

2. Page 2, line 48: I would advise additional keywords: gender discrimination, gender equality, Zambia, representation. And add a common after 'research'.

Response: We have included the suggested key words (page 2) and attended to all the suggested corrections in the document (see tracked changes)

3. Page 5, line 17: There are more up to date numbers from UNESCO here:

<https://www.unesco.org/reports/science/2021/en/dataviz/share-women-researchers-radial> (underlying data is also available).

4. Response: We have included the suggested citation and edited the numbers accordingly

5. Pages 22 to 27: Concerns and corrections on the questionnaire

Response: We have corrected all errors and concerns raised by reviewer on the questionnaire. We have also deleted questions that may looked ambiguous (see tracked changes on the questionnaire)

VERSION 2 – REVIEW

REVIEWER	Coates, Laura Oxford Brookes University Faculty of Health and Life Sciences, NDORMS
REVIEW RETURNED	08-Feb-2023

GENERAL COMMENTS	No further comments
---------------------

REVIEWER	Herbert, Rachel Elsevier BV, International Center for the Study of Research
REVIEW RETURNED	16-Feb-2023

GENERAL COMMENTS	The authors have addressed my comments and concerns satisfactorily on the manuscript. As a result of all of the changes, I perceive that the study goals and approach are a better fit, and that the questionnaire in particular is in better shape and, as such, the results of the analysis will be much more robust and comprehensive. Throughout, there are still minor phrasing and spelling errors. I trust that those can be corrected prior to publication.
---